# Molecular and Kinetic Analyses of Circulating Tumor Cells as Predictive Markers of Treatment Response in Locally Advanced Rectal Cancer Patients

**DOI:** 10.3390/cells8070641

**Published:** 2019-06-26

**Authors:** Bianca C. Troncarelli Flores, Virgilio Souza e Silva, Emne Ali Abdallah, Celso A.L. Mello, Maria Letícia Gobo Silva, Gustavo Gomes Mendes, Alexcia Camila Braun, Samuel Aguiar Junior, Ludmilla Thomé Domingos Chinen

**Affiliations:** 1International Research Center, A.C. Camargo Cancer Center, São Paulo 01508-010, Brazil; 2Department of Medical Oncology, A.C. Camargo Cancer Center, São Paulo 01509-900, Brazil; 3Department of Radiotherapy, A.C. Camargo Cancer Center, São Paulo 01509-900, Brazil; 4Department of Radiology, A.C. Camargo Cancer Center, São Paulo 01509-900, Brazil; 5Department of Pelvic Surgery, A.C. Camargo Cancer Center, São Paulo 01509-900, Brazil; 6National Institute for Science and Technology in Oncogenomics and Therapeutic Innovation, São Paulo 01509-900, Brazil

**Keywords:** locally advanced rectal cancer, circulating tumor cells, RAD23B, thymidylate synthase, chemoradioresistance

## Abstract

Neoadjuvant chemoradiation (NCRT) followed by total mesorectal excision is the standard treatment for locally advanced rectal cancer (LARC). To justify a non-surgical approach, identification of pathologic complete response (pCR) is required. Analysis of circulating tumor cells (CTCs) can be used to evaluate pCR. We hypothesize that monitoring of thymidylate synthase (TYMS) and excision repair protein, RAD23 homolog B (RAD23B), can be used to predict resistance to chemotherapy/radiotherapy. Therefore, the aims of this study were to analyze CTCs from patients with LARC who underwent NCRT plus surgery for expression of TYMS/RAD23B and to evaluate their predictive value. Blood samples from 30 patients were collected prior to NCRT (S1) and prior to surgery (S2). CTCs were isolated and quantified by ISET^®^, proteins were analyzed by immunocytochemistry, and *TYMS* mRNA was detected by chromogenic in situ hybridization. CTC counts decreased between S1 and S2 in patients exhibiting pCR (*p* = 0.02) or partial response (*p* = 0.01). Regarding protein expression, TYMS was absent in 100% of CTCs from patients with pCR (*p* = 0.001) yet was expressed in 83% of non-responders at S2 (*p* < 0.001). Meanwhile, RAD23B was expressed in CTCs from 75% of non-responders at S1 (*p* = 0.01) and in 100% of non-responders at S2 (*p* = 0.001). Surprisingly, 100% of non-responders expressed *TYMS* mRNA at both timepoints (*p* = 0.001). In addition, TYMS/RAD23B was not detected in the CTCs of patients exhibiting pCR (*p* = 0.001). We found 83.3% of sensitivity for TYMS mRNA at S1 (*p* = 0.001) and 100% for TYMS (*p* = 0.064) and RAD23B (*p* = 0.01) protein expression at S2. Thus, *TYMS* mRNA and/or TYMS/RAD23B expression in CTCs, as well as CTC kinetics, have the potential to predict non-response to NCRT and avoid unnecessary radical surgery for LARC patients with pCR.

## 1. Introduction

Colorectal carcinoma is one of the most commonly occurring neoplasms in the Western world. Moreover, despite improvements in treatment, colorectal carcinoma remains an important cause of cancer-related deaths worldwide [1]. Rectal carcinoma, which accounts for approximately 30% of all primary colorectal cancers, is characterized by an anatomy and natural history that are distinct from other colonic tumors [2,3].

Neoadjuvant chemoradiation (NCRT) followed by total mesorectal excision is the standard treatment for locally advanced rectal carcinoma (LARC). NCRT is generally recommended for patients with cT3/T4 disease or lymph node involvement [4].

The absence of viable tumor cells in surgical specimens after NCRT is defined as pathological complete response (pCR). It occurs in 10–30% of patients [5] and it is associated with a better prognosis. Capecitabin and 5-fluorouracil (5-FU) are currently the most widely used radiosensitizers for NCRT. In some patients, chemoradiation induces pCR, although radical surgery is recommended for most patients [6,7].

Accurate identification of patients with pCR is necessary before pursuing a non-surgical approach. Many studies have evaluated the accuracy of the digital rectal exam and images in identifying patients with pCR [8,9]. However, the sensitivity and specificity of these methods are too low to accurately predict pCR. Molecular analyses of gene expression have also been performed and have been shown to be unsuitable for identifying patients with pCR who can be followed without radical surgery [10]. Recently, promising data have supported the use of a “watch and wait” strategy for patients who have no signs of a viable tumor in a digital rectal exam, rectoscopy (with or without biopsy), or imaging. The likelihood of tumor regrowth is minimal for these patients, and most could be cured by salvage surgery [11].

In addition, considering that the majority of patients will not respond to NCRT, it is extremely important to identify these patients, in an attempt to optimize the response to these patients. As there are no randomized phase-III studies to date, comparing this strategy with isolated induction chemoradiotherapy, it becomes fundamental to identify a biomarker capable of selecting patients for these approach [12,13,14].

For organ-preserving treatment strategies, the ability to identify tissues or blood biomarkers that predict NCRT response/non-response is very important. Circulating tumor cells (CTCs) are a real-time source of biomarkers, which have shown promise in facilitating the detection and monitoring of pCR and non-responder patients. It has been proposed that the identification and analysis of CTCs would facilitate investigations to understand intrinsic tumor features and characteristics of patients with pCR or non-responders so that individualized treatment strategies can be applied [15]. To date, use of CTC counts has been approved by the US Food and Drug Administration as a prognostic tool for metastatic prostate, colon, and breast cancers [16,17,18]. Additionally, CTC kinetics can be used to monitor tumor response to systemic treatment [19,20]. Meanwhile, molecular characterization of CTCs has facilitated studies of biomarkers, including proteins, gene expression, and chromosomal translocations [21,22].

Since only a small proportion of patients with LARC experience complete response following chemoradiation, an improved understanding of the molecular mechanisms underlying resistance to chemotherapy and radiation therapy resistance is essential. For chemotherapy, the main target of 5-FU is the enzyme thymidylate synthase (TYMS). TYMS expression analysis has been used to predict individual response to NCRT and has exhibited good prognostic value for rectal cancer recurrence [23,24]. Radiation therapy is an effective component of neoadjuvant treatment and it induces genetic damage. Consequently, the ultraviolet excision repair protein, RAD23 homolog B (RAD23B), which is part of the nucleotide excision repair process [25,26], would potentially be induced by the genetic damage introduced by radiation therapy. Recently, RAD23B was found to be associated with breast cancer relapse risk [27].

The aim of the present study was to explore the role of CTCs in patients undergoing NCRT followed by surgery for treatment of LARC. In addition, the predictive values of TYMS and RAD23B before and after NCRT were evaluated.

## 2. Methods

### 2.1. Patients and Treatment

This prospective study was conducted at the A. C. Camargo Cancer Center (São Paulo, Brazil) and was approved by the local ethics committee (2141/15C). Written informed consent was obtained from all patients prior to enrolment. Patients met the inclusion criteria for this study if they had a diagnosis of rectal cancer, as confirmed by biopsy pathology; had locally advanced disease staged as cT3–cT4 or N0–N+; and were candidates for chemoradiation therapy per institutional protocol. Patients were excluded if they had evidence of distant metastasis; a history of any surgery (e.g., colostomy) within two weeks prior to the initiation of treatment; or were taking anticoagulants at the time of the study. Cancer stage was determined with pelvic magnetic resonance imaging and chest and abdominal computed tomography.

Blood samples were collected at baseline or prior to NCRT (S1), and then prior to surgery (S2). Venous blood was collected from the antecubital vein and these samples were stored at room temperature for a maximum of 4 h prior to analysis.

Radiation therapy was applied with a three-dimensional (3D) conformal technique. A 45-Gy dose was applied in 25 fractions over the entire pelvis. In addition, a 5.4-Gy radiation boost was administered to the primary tumor and involved lymph nodes in three fractions, for a total of 50.4 Gy applied in 28 fractions. Chemotherapy regimens consisted of either intravenous 5-FU administered at a dose of 1000 mg/m^2^ on days 1–5 during weeks 1 and 5 of radiation therapy; or oral capecitabin administered at a dose of 1650 mg/m^2^/d during the entire radiation treatment period. Each patient’s physician determined which regimen was appropriate.

The evaluation of the response was determined by the comparative analysis of the baseline images in S1 with the images before surgery (S2). In addition, we evaluated the pathologic response in comparison to the clinical staging established by baseline images.

### 2.2. Isolation and Quantification of CTCs and Protein Analysis of TYMS and RAD 

ISET^®^ (Isolation by size of epithelial tumors) was used to quantify and analyze CTCs. Briefly, peripheral blood samples were collected from patients into EDTA tubes (8.0 mL BD Vacutainer, Franklin Lakes, NJ, USA) and then were homogenized at room temperature for up to 4 h. Samples were then prepared as described previously [28]. Briefly, ISET membrane spots were cut out and subjected to immunocytochemistry (ICC) with an anti-RAD23B antibody (1:100 CSB-PA019260LA01HU; CusaBio, Wuhan, People’s Republic of China), an anti-TYMS antibody (1:230 WH0007298M1; Sigma-Aldrich, St. Louis, MO, USA), and an anti-CD45 antibody (1:200 HPA000440; Sigma-Aldrich).

Selected spots from the ISET membranes were additionally subjected to a 24-well dual color ICC assay (Polink DS-RR-Hu/Ms A Kit; GBI Labs, Bothell, WA, USA). Briefly, antigen retrieval was performed by using a DakoTM Antigen Retrieval Solution (Dako, Santa Clara, CA, USA). Cells were then hydrated with 1X Tris-buffered saline (TBS) for 10 min and then permeabilized with Triton X-100 for 5 min. Next, cells were rinsed with 1X TBS and incubated with 3% hydrogen peroxide in the dark for 15 min to block endogenous peroxides. Immobilized cells on the membrane spots were incubated overnight with primary antibodies diluted in 10% fetal calf serum in TBS. To amplify primary antibody signals, the spots were incubated for 30 min with rabbit horseradish peroxidase (HRP) polymer (GBI Labs), then with 3,3-diaminobenzidine (DAB) for 10 min. After amplification, the spots were incubated with a second primary antibody for 2 h, a rabbit AP polymer (GBI Labs) for 30 min, and then GBI-Permanent Red (GBI Labs) for 10 min. The latter reagent was freshly prepared according to the manufacturer’s instructions. After staining the cells with haematoxylin, specimens were examined with light microscopy (Research System Microscope BX61; Olympus, Waltham, MA, USA). CTCs were counted per 1 mL blood, as previously described by Krebs et al. [29]. CTCs were characterized according to five criteria: Negativity for CD45 staining; nucleus size >12 µm, hyperchromatic and irregular nuclei; visible cytoplasm; and a nuclear to cytoplasm ratio >80% [30]. CTCs were considered positive for TYMS or RAD23B expression if at least one cell in a specimen stained for these markers on ICC analysis.

### 2.3. CTC Isolation and Immunostaining Control

Negative controls were healthy donor blood filtered by ISET^®^. Positive controls included healthy donor blood spiked with HCT-8 colorectal carcinoma cells. For the ICC reaction and TYMS antibody control, leukocytes from healthy filtered blood were used. According to the Human Protein Atlas (http://www.proteinatlas.org/), the latter express TYMS. As a positive control for the RAD23B antibody, HCT-8 cells were spiked in healthy donor blood and filtered on ISET^®^. To create a negative control for ICC, the same cell line was used without primary antibodies in order to ensure exclusion of cross-reactivity. To confirm that analyzed CTCs were not leukocytes, staining with an anti-CD45 antibody was performed. 

### 2.4. Chromogenic In situ Hybridization Assay for TYMS

*TYMS* mRNA was detected in intact cells by using a chromogenic in situ hybridization (CISH) assay employing RNAscope Technology (ACDbio, Newark, CA, USA), according to the manufacturer’s protocol. Cytology methods standardized by our group were also used. Briefly, one spot was cut out for each patient sample and these were placed in individual wells of a 24-well plate. The membrane spots were hydrated with 1X TBS for 5 min before being incubated in 1% formaldehyde solution for 5 min at room temperature. The spots were then rinsed 2× with distilled water before applying 5–8 drops of RNAscope Hydrogen Peroxide (ACDbio) to each spot. After incubating the samples in a humid chamber for 10 min at room temperature, the spots were washed 2× with distilled water and mounted on slides. To each slide, 3 drops of cytology pepsin were applied. After 10 min at room temperature the spots were returned to the 24-well plate, washed 2× with distilled water, then rapidly dehydrated in successive 1-min incubations with 70%, 85%, and 100% ethanol solutions. After the spots were left to air dry on the slides, 3 drops of a TYMS-specific probe were added to each spot. After 2 h at 40 °C in a HybEZ oven hybridizer (ACDbio), drops of Amp1-6 solutions were added, according to the manufacturer’s protocol, with washes performed with 1X TBS. Finally, the cells were incubated with chromogen and 50% haematoxylin then placed on slides with aqueous mounting media and coverslips. The samples were inspected with brightfield microscopy (BX61-Olympus; Olympus). To classify *TYMS* mRNA expression, an absence or presence of staining was classified as negative or positive, respectively.

### 2.5. Statistical Analysis

Initially, a descriptive analysis was performed to obtain absolute (*n*) and relative (%) frequency distributions. To evaluate possible associations between the variables of interest, a contingency table was constructed from sample data. Chi-square tests of independence or Fisher’s exact test were used, as appropriate. The level of significance was set at 5% and all statistical analyses were performed by using the SPSS program for Windows, version 25.

## 3. Results

### 3.1. Patients

Thirty patients with rectal cancer were enrolled in this study between April 2016 and January 2018. Clinical characteristics of these patients are listed in Table 1. The mean age of this cohort was 56 years (range, 34–72) and 60% of the patients were male. For 67% of the patients, their rectal tumors were located 7 cm or less from the anal verge.

The baseline (prior to NCRT) T stage was cT2 for 4 patients (13%), cT3 for 21 patients (70%), and cT4 for 5 patients (17%). Baseline N stage was cN0 for 22 patients (73%) and cN+ for 8 patients (27%). The mean time to surgery completion following NCRT was 77 days. Pathologic T stage was pT0 for 6 patients (20%), pT1 for 5 patients (17%), pT2 for 7 patients (23%), pT3 for 9 patients (30%), and pT4 for 2 patients (7%). Pathologic N stage was pN0 for 21 patients (70%), pN1 for 6 patients (20%), and pN2 for 2 patients (7%). One patient (3%) exhibited disease progression during NCRT. Complete pathologic response after preoperative therapy was detected in 6 patients (20%), while 7 patients (23%) had their tumors down-staged (ypT1-2N0).

All patients were treated with 25 fractions of 45 Gy to the pelvis with a 3D conformal technique. In addition, a 5.4-Gy radiation boost was applied to the primary tumor. There were no treatment delays or interruptions that lasted more than two days. Twenty-six patients (86.7%) received intravenous 5-FU and four patients (13.3%) received capecitabine. The main treatment toxicities were grade 1 or 2 diarrhea in 10 patients (33%) and grade 1 or 2 oral mucositis in 8 patients (27%). No grade 3 or 4 toxicities were reported. There were also no differences in adverse events between the two chemotherapy groups.

### 3.2. CTCs

CTCs were detected at S1 in all 30 patients of our cohort, and at S2 in 24 patients (Figure 1A,D). The mean CTC concentrations were 6 cells/mL at S1 and 3.5 cells/mL at S2. Kinetic analyses showed that CTC levels were increased in 3 patients, decreased in 22 patients, and remained unchanged in 5 patients. The mean CTC count per mL was 3.1 for those with pCR, 2.5 for those exhibiting a partial response (PR), and 2.9 for those with non-responsive tumors. Patients exhibiting pCR and PR showed a decrease in CTC kinetics (calculated as: CTC baseline [CTC1] × CTC post-CRT [CTC2]) during treatment (*P* = 0.02 and *P* = 0.01, respectively; Table 2).

### 3.3. TYMS and RAD23B

At baseline, TYMS-positive CTCs were detected in 7 patients (23.5%) by ICC (Figure 1B) and in 21 patients (70%) by CISH (Figure 1E,F). RAD23B-positive CTCs were detected in 13 patients (43.3%) by ICC (Figure 1C).

After chemoradiation, TYMS-positive CTCs were detected in 10 patients (41.6%) by ICC, while *TYMS* mRNA was detected in 16 patients (61.5%) by CISH. RAD23B-positive CTCs were detected in 14 patients (58.3%).

Baseline *TYMS* mRNA and RAD23B-positive CTCs were associated with poor clinical response. For example, all 12 patients with non-responsive tumors had *TYMS* mRNA detected in their CTCs by CISH. In contrast, 83.5% of patients with pCR did not have CTCs expressing *TYMS* mRNA (*P* = 0.001; Table 3). At S1, RAD23B-positive CTCs were detected in 33% of patients with pCR, in 16% of patients with PR, and in 75% of patients exhibiting no response (*P* = 0.01). Thus, there was no association between detection of TYMS expression in CTCs by ICC and response type in S1 (Table 3). To confirm the value of these proteins as biomarkers, we compared these found with the most used clinical parameter, which is pathological response of the primary tumor. We found, for TYMS mRNA, 83.3% sensitivity, 83.3% specificity, and 95.2% positive predictive value (PPV) (*p* = 0.001) (Appendix A).

Post-chemoradiation analyses (S2) showed that expression of TYMS and RAD23B in CTCs strongly correlates with poor response. For example, TYMS-positive CTCs were not detected in the patients with pCR or pPR, yet they were detected in 83% of non-responsive patients (*P* < 0.001; Table 3). Furthermore, *TYMS* mRNA expression at S2 correlated with response, with *TYMS* mRNA detected in all of the patients exhibiting no response (*P* = 0.001). We found, for TYMS protein, 100% sensitivity, 50% specificity, and 100% positive predictive value (PPV) (*p* = 0.064). For RAD23B, the values were: 100% sensitivity, 70% specificity, and 100% positive predictive value (PPV) (*p* = 0.01) (Appendix A).

Lastly, we found that expression of TYMS and RAD23B in CTCs was strongly predictive of response type. For example, after NCRT, TYMS^−^/RAD23B^−^ CTCs were detected in 100% of the pCR patients, in 83.5% of the PR patients, and in none of the NR patients. However, TYMS^+^/RAD23B^+^ CTCs were not detected in patients with pCR or PR, and yet were detected in 83.5% of patients with no response (*P* = 0.001). At S1, CTCs expressing TYMS and RAD23B did not correlate with response type (*P* = 0.1; Table 4). Meanwhile, expression of *TYMS* mRNA determined which patients did not respond to chemoradiotherapy at S1 and S2 (Figure 2).

## 4. Discussion

LARC treatment generally consists of NCRT followed by total mesorectal excision. For high-risk patients, postoperative adjuvant chemotherapy may additionally be considered [31]. The ability to identify patients who have undergone complete eradication of a tumor following NCRT is crucial. In the current study, we prospectively demonstrated a strong correlation between expression of TYMS and RAD23B by CTCs in patients with LARC and pCR following NCRT.

Among the 50–60% of patients with LARC who respond to NCRT (i.e., their tumors are down-staged following treatment), many exhibit improved survival. It has been reported that pCR after NCRT is associated with improved cancer outcome and significantly decreased rates of local recurrence. Conversely, the 40% non-response rate after NCRT [31] represents heterogeneity in response to standard treatment [32,33]. For the latter, new therapeutic strategies, avoidance of toxicity associated with ineffective treatment, and novel neoadjuvant chemotherapeutic options are needed [34,35]. In addition, the identification of biomarkers would facilitate the creation of individualized treatment plans.

The ability to identify tumors that do not respond to radiotherapy is useful for helping patients avoid radiation side effects such as fibrosis, fecal and bladder incontinence, diarrhea, dysuria, and myelosuppression [36,37]. This knowledge would also facilitate discussions regarding alternate approaches, such as more potent neoadjuvant multi-agent chemotherapy strategies or a rationale for foregoing radiation therapy. There are several studies that have discussed these considerations, although robust biomarkers that will predict non-responders of NCRT with high accuracy still need to be identified [37,38,39]. In the present study, we were able to identify NCRT non-responders at S1 by detecting *TYMS* mRNA. Thus, it is possible that detection of *TYMS* mRNA in CTCs could represent a valuable tool in identifying non-responder patients prior to the start of NCRT. Furthermore, detection of RAD23B expression could make this patient selection process more accurate. It is well-established that RAD23B is involved in DNA repair following radiation damage, and its identification in CTCs can guide treatment plans. In the present study, 75% of non-responding tumors expressed this protein on CTCs before NCRT. Furthermore, when both RAD23B and TYMS protein expression on CTCs were detected after NCRT, 100% of patients with pCR did not express either protein. Meanwhile, 83.5% of patients with non-responsive tumors expressed both proteins and the remaining 16.5% expressed at least one of these proteins. In addition, *TYMS* mRNA expression after NCRT showed high positivity for non-responders (100%) and was not related to protein expression (Figure 3). For the seven patients who presented discordant mRNA/protein positivity at S1 (Appendix A), they exhibited a correspondence between *TYMS* gene expression and TYMS protein expression at S2. Chemotherapeutic agents have previously been associated with changes in gene expression. In addition, post-transcriptional mechanisms for blocking protein synthesis have been characterized [40,41,42]. For 5-FU, there are several papers that describe this correlation [43,44]. The present results support the hypothesis that CTC analysis can be a useful tool for identifying patients who will respond to chemoradiotherapy. As a result, a “watch and wait” strategy becomes an option to be considered in addition to radical surgery.

Our analysis of RAD23B and TYMS expression showed similar profiles for these two proteins. Based on the different protein patterns for TYMS in relation to pCR and PR at S1 and S2, we decided to further examine the mRNA expression of *TYMS* with RNA hybridization assays performed in situ with CTCs. All of the CTCs from non-responding patients expressed *TYMS* mRNA at both S1 and S2. Thus, this assay exhibited high sensitivity and specificity for identifying LARC patients who are predicted to be non-responders to radiochemotherapy.

After NCRT, TYMS expression exhibited a strong correlation with *TYMS* mRNA (kappa value 0.6; *P* = 0.003) in absence of response to treatment. This second assay at S2 cost less to perform besides the possibility of performing more than one protein per time on the CTCs isolated by ISET^®^ methodology. Furthermore, we previously showed that elevated TYMS expression in CTCs was associated with poor prognosis among patients with metastatic colorectal cancer [26].

TYMS, a downstream target molecule of 5-FU, plays a key role in DNA synthesis. The enzyme catalyzes deoxyuridine monophosphate methylation to produce deoxythymidine triphosphate, and subsequently, thymidylate. Increased TYMS expression is thought to be responsible for 5-FU resistance in patients with colorectal cancer [42,43]. Meanwhile, RAD23B is a member of the nucleotide excision repair system, which stabilizes the xeroderma pigmentosum complementation group C (XPC) protein and potentiates its interaction with damaged DNA [45]. XPC subsequently initiates nucleotide excision repair. It is not clear why increased production of RAD23B protein was observed in CTCs from non-responders to NCRT in the present study. A possible hypothesis is that increased levels of this repair protein during treatment make it difficult to eliminate cancer cells by NCRT, a strategy that is based on inducing apoptosis as a result of DNA damage.

A treatment approach that deserves discussion in the context of our study is the “watch and wait” strategy. For the patients whose tumors respond completely to NCRT (as evidenced by radiologic, clinical, and endoscopic evaluations), a “watch and wait” strategy may allow patients to avoid the morbidity, mortality, and functional consequences of radical surgical treatment. A recently published meta-analysis of 880 patients who were monitored with a "watch and wait" strategy showed that the strategy can be safely incorporated into a multidisciplinary management plan for the treatment of patients with rectal cancer who achieve pCR after neoadjuvant treatment [12,46,47,48]. Therefore, identification of a biomarker that can predict pCR to NCRT treatment prior to treatment, or in the early stages of treatment, would be of great clinical utility in combination with commonly used clinical parameters.

In the present study, it was observed that 100% of patients who responded to NCRT did not express TYMS or RAD23B proteins on their CTCs at the beginning of their follow-up monitoring. This lack of protein expression in CTCs is consistent with imaging and kinetic studies that have showed a reduction or elimination of CTCs post NCRT. Furthermore, we observed that CTC kinetics correlated with disease outcome for our patients with LARC. To the best of our knowledge, this is the first study to demonstrate this result. The observed decrease in CTC levels during treatment in our cases involving complete or partial responses further support the use of CTC analyses to predict response to NCRT.

In summary, our results provide valuable data regarding two potential biomarkers of chemoradiation resistance in patients undergoing neoadjuvant treatment for LARC. Despite the small sample size of our study, CTC kinetics, as well as *TYMS* mRNA and/or RAD23B/TYMS protein expression in CTCs, were found to strongly correlate with pCR. Further studies are needed to validate these findings with a larger patient cohort. If CTC analysis proves useful in predicting pCR with high accuracy, many patients may be spared radical surgery for rectal cancer treatment. In addition, biomarker and kinetic analyses of CTCs may identify potential non-responders to treatment with NCRT, thereby identifying a need to evaluate other forms of therapy for these patients.

## Figures and Tables

**Figure 1 cells-08-00641-f001:**
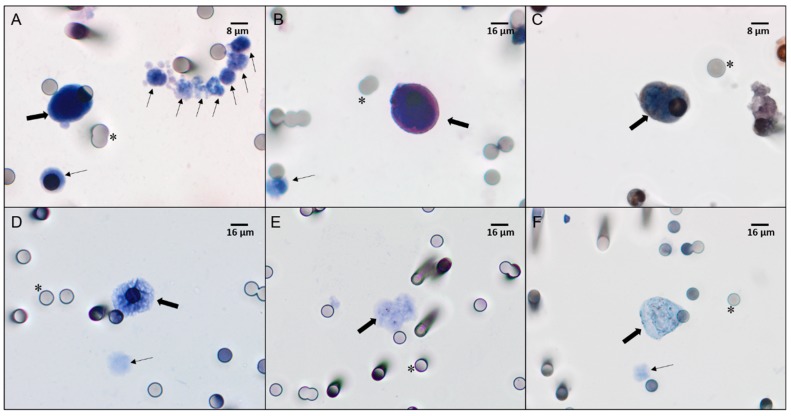
Immunostaining (**A–C**) and chromogenic in situ hybridization (CISH) (**D–F**) of CTCs from locally advanced rectal cancer (LARC) patients. (**A**) CTCs and leukocytes visualized with haematoxylin-eosin staining (×40 magnification). (**B**) CTCs stained with an anti-thymidylate synthase (TYMS) antibody, visualized with Permanent Red, and counterstained with haematoxylin (×40 magnification). (**C**) CTCs stained with an anti-RAD23B antibody, visualized with 3,3-diaminobenzidine (DAB), and counterstained with haematoxylin (×60 magnification). (**D**) CTCs negative for *TYMS* mRNA and counterstained with haematoxylin (×40 magnification). (**E**) CTCs with a low *TYMS* mRNA signal (normal expression) counterstained with haematoxylin (×40 magnification). (**F**) CTCs with a high *TYMS* mRNA signal (overexpression) counterstained with haematoxylin (×40 magnification). All images were analyzed on a Research System Microscope BX61 (Olympus, Tokyo, Japan) coupled to a digital camera (SC100–Olympus). Thick arrows indicate CTCs, thin arrows indicate leukocytes, and asterisks indicate 8 μm pores of the ISET^®^ membranes.

**Figure 2 cells-08-00641-f002:**
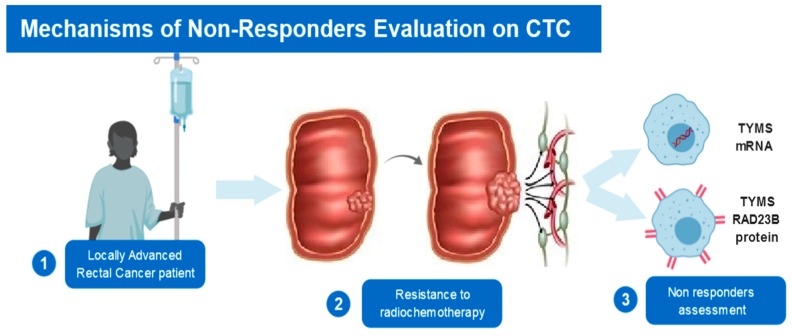
Scheme demonstrating both methodologies used to select responders and non-responders to neoadjuvant chemoradiotherapy: mRNA detected by CISH and protein expression detected by immunocytochemistry (ICC).

**Figure 3 cells-08-00641-f003:**
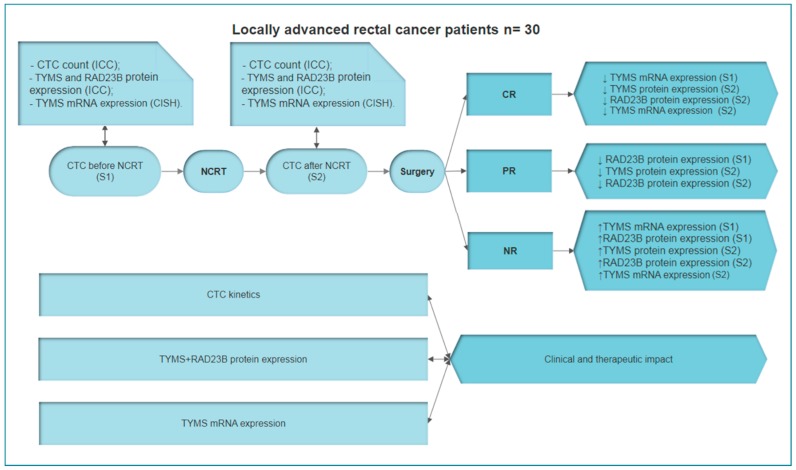
Summary of methodologies, analyses, and results in the present study. Patients were enrolled prior to the start of NCRT. Blood was collected to perform CTC counts and molecular analyses. At baseline, complete response (CR) correlated with low levels of *TYMS* mRNA. In contrast, NR correlated with high levels of *TYMS* mRNA and RAD23B protein expression. Blood samples were collected again during follow-up after NCRT. CTC analyses showed that CR correlated to low levels of *TYMS* mRNA and RAD23B protein expression, while NR correlated to high levels of *TYMS* mRNA and RAD23B/TYMS protein expression. CTC kinetics also correlated to pathological response. Based on these results, *TYMS* mRNA and RAD23B and TYMS protein expression appear to have a clinical and therapeutic impact in LARC patients. Abbreviations: CR: Complete response; LARC: Locally advanced rectal cancer; NCRT: Neoadjuvant chemoradiotherapy; NR: No response; PR: Partial response.

**Table 1 cells-08-00641-t001:** Patient characteristics.

Characteristics	*N (%)*
Average age (min–max), years	56 (34–72)
Gender	
Male	18 (60)
Female	12 (40)
Tumor distance from the anal verge	
≤7 cm	20 (67)
>7 cm	10 (33)
Clinical T baseline stage	
T2	04 (13)
T3	21 (70)
T4	05 (17)
Clinical N baseline stage	
N0	22 (73)
N+	08 (27)
Pathological T stage	
T0	06 (20)
T1–T2	12 (40)
T3–T4	11 (37)
DP	1 (3)
Pathological N stage	
N0	21 (70)
N1–N2	08 (27)
DP	01 (3)
Average time (min–max) of completion of RDT for surgery (days)	77 (50–143)

Abbreviations: DP: Disease Progression; RDT: radiotherapy.

**Table 2 cells-08-00641-t002:** Kinetic counts of circulating tumor cells (CTCs) between baseline (CTC1) and post-neoadjuvant chemoradiation (NCRT) (CTC2) time points.

9	Patient ID	CTCs/mL before NCRT	CTCs/mL after NCRT	Kinetics of CTC1 vs. CTC2	
**CR**	8	3	1	>	*P* = 0.02
	11	4	1	>	
	18	1	0	>	
	21	4	0	>	
	25	4	2	>	
	27	3	2	>	
**PR**	3	5	5	=	*P* = 0.01
	4	3	2	>	
	7	3	2	>	
	9	1	0	>	
	10	0	1	<	
	13	1	1	=	
	15	2	0	>	
	16	2	0	>	
	23	2	1	>	
	24	6	3	>	
	29	3	0	>	
	30	2	2	=	
**NR**	1	3	4	<	*P* = 0.07
	2	3	2	>	
	5	1	1	=	
	6	1	1	=	
	12	1	4	<	
	14	2	1	>	
	17	3	1	>	
	19	4	2	>	
	20	5	1	>	
	22	8	4	>	
	26	2	1	>	
	28	2	1	>	

Abbreviations: NCRT: Neoadjuvant chemoradiotherapy; CR: Complete response; PR: Partial response; NR: No response.

**Table 3 cells-08-00641-t003:** Expression profiles of RAD23B and TYMS proteins and *TYMS* mRNA.

			*CR*	*PR*	*NR*	
Before NCRT	CISH (TYMS)	+	16.5	66.5	100	*P* = 0.001
		−	83.5	33.5	0	
	TYMS (protein)	+	16.5	25	25	*P* = 1.0
		−	83.5	75	75	
	RAD (protein)	+	33.5	16.5	75	*P* = 0.01
		−	66.5	83.5	25	
After NCRT	CISH (TYMS)	+	25	30	100	*P* = 0.001
		−	75	70	0	
	TYMS (protein)	+	0	0	83.5	*P* = 0.0001
		−	100	100	16.5	
	RAD (protein)	+	0	25	100	*P* = 0.0001
		−	100	75	0	

Abbreviations: NCRT: Neoadjuvant chemoradiotherapy; CR: Complete response; PR: Partial response; NR: No response.

**Table 4 cells-08-00641-t004:** Correlation between RAD and TYMS protein expression profiles.

Profile	*CR %*	*PR %*	*NR %*	
TYMS−/RAD−	50	66.5	16.5	Before NCRT *P* = 0.1
TYMS+/RAD+	0	8.5	16.5	
TYMS+/RAD− TYMS−/RAD+	50	25	67	
TYMS−/RAD−	100	83.5	0	After NCRT *P* = 0.001
TYMS+/RAD+	0	0	83.5	
TYMS+/RAD− TYMS−/RAD+	0	16.5	16.5	

Abbreviations: NCRT: Neoadjuvant chemoradiotherapy; CR: Complete response; PR: Partial response; NR: No response.

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
