# Peer review of "Molecular and Kinetic Analyses of Circulating Tumor Cells as Predictive Markers of Treatment Response in Locally Advanced Rectal Cancer Patients"

_cells, 2019, doi:10.3390/cells8070641_

Round 1

Reviewer 1 Report

This paper is about looking at circulating tumor cells thymidylate synthase and excision repair protein rad23 homolog b, which can be used to predict resistance to chemotherapy. 1. Normally histopathologic specimens are evaluated by submitting the entire tumor bed in rectal carcinomas after neoadjuvant therapy to look for pathologic response. Predicting histopathologic response is very subjective amongst pathologist due to lot of interpersonal variation. In this paper they used the above 2 mentioned markers to predict response to therapy and state if they are chemo resistant or sensitive.  2. In this paper they looked at 30 patients with locally advanced rectal carcinoma and collected blood samples before and after neoadjuvant therapy.  3. Patients who expressed baseline TYMS mRNA and RAD23B CTC were associated with poor clinical response. All non-responders expressed TYMS mRNA CTC by CISH. 4. Patients with pCR and pPR had no expression of TYMS mRNA in CTC thus predicting good response.  5. TYMS mRNA and RAD23B CTC strongly predicted response type. Strengths of the paper 1. Very well written with a clear concept. 2. Material and methods clearly describe how they went about achieving the results. 3. Both markers are potent biomarkers that if present predict resistance to therapy even before NCRT. 4. 100% of patients with pCR did not express both the CTC. 5. CTC kinetics correlation with patients outcome.

Author Response

Ms. Milena Amidzic,

Assistant Editor, MDPI Belgrade Office

Cells

Ref. Manuscript: cells-513256

MS: " Molecular and Kinetic Analyses of Circulating Tumor Cells as Predictive Markers of Treatment Response in Locally Advanced Rectal Cancer Patients".

Dear Ms. Milena Amidzic,

Thank you so much for your email from June 3rd, 2019 and for sharing with us the positive evaluation of our manuscript by your referees. New/modified text in the manuscript is uploaded, with the English review asked by your referees. We decided to modify the title of the manuscript to: “Molecular and Kinetic Analyses of Circulating Tumor Cells as Predictive Markers of Treatment Response in Locally Advanced Rectal Cancer Patients” and also included a supplementary table. In this point-by-point reply we address the points raised by reviewer two, as follows:

Reviewer 1:

This paper is about looking at circulating tumor cells thymidylate synthase and excision repair protein rad23 homolog b, which can be used to predict resistance to chemotherapy. 1. Normally histopathologic specimens are evaluated by submitting the entire tumor bed in rectal carcinomas after neoadjuvant therapy to look for pathologic response. Predicting histopathologic response 

is very subjective amongst pathologist due to lot of interpersonal variation. In this paper they used the above 2 mentioned markers to predict response to therapy and state if they are chemo resistant or 

sensitive. 

2. In this paper they looked at 30 patients with locally advanced rectal carcinoma and collected blood

samples before and after neoadjuvant therapy. 

3. Patients who expressed baseline TYMS mRNA and RAD23B CTC were associated with poor clinical response. All non-responders expressed TYMS mRNA CTC by CISH. 4. Patients with pCR and pPR had no expression of TYMS mRNA in CTC thus predicting good response.  5. TYMS mRNA and RAD23B CTC strongly predicted response type. Strengths of the paper 1. Very well written with a clear concept. 2. Material and methods clearly describe how they went about achieving the results. 3. Both markers are potent biomarkers that if present predict resistance to therapy even before NCRT.

4. 100% of patients with pCR did not express both the CTC. 5. CTC kinetics correlation with patients outcome.

Answer to comment 1:

We thank Referee 1 for his kind words, and for considering our paper for publication. Thank you very much!

Reviewer 2:

This manuscript presents the results of a study focusing on a topic of really high interest. The conclusions derived are highly intriguing and shed light in one of the least understood aspects of the biology of rectal cancer. The clinical implications are of sustantial importance. I have minor remarks for the authorsQ

The introduction may be minimised to the most relevant information of the problem of non-response and the efforts done up to now to resolve it

Although predictive values are included in the results, I would suggest to add sensitivity and specificity values for the measured parameters as they may be more clinicalllly comprehensible

Answer to comment 2:

We thank this referee for the careful analysis of our manuscript and for their useful suggestions

Answer to comment 1: We minimized the introduction and pointed the most relevant information, of the problem of non-response and the efforts done up to now to resolve it. Thank you very much for pointing this. Your suggestion made our paper more clear.

Answer to comment 2:  We thank Reviewer #2 for this useful suggestion. We made the analyses suggested and we agree with you. The additional analysis made our data more clinically comprehensible. Thank you very much for the insightful observation that helped us to improve the manuscript.

Reviewer 2 Report

This manuscript presents the results of a study focusing on a topic of really high interest. The conclusions derived are highly intriguing and shed light in one of the least understood aspects of the biology of rectal cancer. The clinical implications are of sustantial importance. I have minor remarks for the authorsQ

The introduction may be minimised to the most relevant information of the problem of non-response and the efforts done up to now to resolve it

Although predictive values are included in the results, I would suggest to add sensitivity and specificity values for the measured parameters as they may be more clinicalllly comprehensible

Author Response

(The authors gave the same response as above.)
